# COVID-19 Bivalent Booster in Pregnancy: Maternal and Neonatal Antibody Response to Omicron BA.5, BQ.1, BF.7 and XBB.1.5 SARS-CoV-2

**DOI:** 10.3390/vaccines11091425

**Published:** 2023-08-28

**Authors:** Wei-Chun Chen, Shu-Yu Hu, Ching-Fen Shen, Hui-Yu Chuang, Chin-Ru Ker, Ching-Ju Shen, Chao-Min Cheng

**Affiliations:** 1Institute of Biomedical Engineering, National Tsing Hua University, Hsinchu 300, Taiwan; lionsmanic@gmail.com (W.-C.C.); chloe.natshuun@gmail.com (S.-Y.H.); 2Division of Gynecologic Oncology, Department of Obstetrics and Gynecology, Chang Gung Memorial Hospital at Linkou, College of Medicine, Chang Gung University, Taoyuan 333, Taiwan; 3Department of Obstetrics and Gynecology, New Taipei City Municipal Tucheng Hospital, New Taipei City 236, Taiwan; 4International Intercollegiate Ph.D. Program, National Tsing Hua University, Hsinchu 300, Taiwan; 5School of Traditional Chinese Medicine, Chang Gung University, Taoyuan 333, Taiwan; 6Department of Pediatrics, National Cheng Kung University Hospital, College of Medicine, National Cheng Kung University, Tainan 701, Taiwan; drshen1112@gmail.com; 7Department of Obstetrics and Gynecology, Kaohsiung Medical University Hospital, Kaohsiung Medical University, Kaohsiung 807, Taiwan; kawaiifish0517@yahoo.com.tw (H.-Y.C.); ruruk19@hotmail.com (C.-R.K.); 8Graduate Institute of Clinical Medicine, College of Medicine, Kaohsiung Medical University, Kaohsiung 807, Taiwan

**Keywords:** bivalent COVID-19 booster vaccines, SARS-CoV-2, Omicron subvariants, BA.5, BF.7, BQ.1, XBB.1.5

## Abstract

Our study was to investigate the effects of bivalent COVID-19 booster vaccination during pregnancy on neutralizing antibody (Nab) levels in maternal blood (MB), transplacental transmission in umbilical cord blood (CB), and efficacy against Omicron SARS-CoV-2 subvariants including BA.5, BF.7, BQ.1, and XBB.1.5. We collected MB and CB from 11 pregnant participants during baby delivery and detected Nab inhibition by enzyme-linked immunosorbent assays (ELISA). Nab inhibition was 89–94% in MB and 82–89% in CB for Omicron subvariants. Those receiving AZD1222 vaccines in previous monovalent vaccination demonstrated poorer maternal Nab inhibition of BA.5, BQ.1, and XBB.1.5 than others. Poorer maternal Nab inhibition of BA.5, BF.7, and BQ.1 was found in those receiving two-dose AZD1222 vaccinations than with either one or no AZD1222 vaccination. MB from those with infants weighing < 3100 g demonstrated better Nab inhibition of BF.7 than those > 3100 g (97.99 vs. 95.20%, *p* = 0.048), and there were also similar trends for Nab inhibition of BA.5 and BQ.1. No significant differences were seen in CB samples. Although diminished maternal Nab inhibition was seen in those with previous monovalent AZD1222 vaccination and heavier newborns, neonatal Nab inhibition was still strong after bivalent COVID-19 booster vaccination.

## 1. Introduction

Since 2019, COVID-19, due to SARS-CoV-2, has created an unprecedented global health crisis [1]. Its rapid spread and high transmission rate have underscored the necessity for effective countermeasures. The development and dissemination of vaccines have been pivotal in slowing the disease’s progression, notably minimizing severe illness among at-risk groups such as healthcare workers, the elderly, and pregnant women [2,3]. COVID-19 presents enhanced risks to pregnant women, including a heightened susceptibility to severe disease and complications such as preeclampsia, preterm birth, and emergent Cesarean delivery [4]. Furthermore, neonates from COVID-19-positive mothers face a greater likelihood of infection, the need for specialized care, and extended hospital stays post-birth [5,6].

Other studies, including our own previously published research, have demonstrated the importance of transplacental neutralizing antibody (Nab) transmission from mothers vaccinated against COVID-19, or other diseases such as influenza, to their babies [7,8,9,10,11,12]. Our previous data demonstrated cord-to-maternal Nab inhibition rate ratios of 0.99 and 0.90 for SARS-CoV-2 wildtype and Delta variants (B.1.617.2) following receipt of two doses of COVID-19 vaccine, respectively [12]. The transmission of Nab inhibition provides newborns with immunity against diseases, including SARS-CoV-2. In a similar transmission phenomenon, fetuses can acquire antibodies from infected mothers, emphasizing the critical need for pregnant women to receive vaccinations for broadened protection of newborns. Our previous research demonstrated that the protective effect of the COVID-19 vaccine is not affected by vaccination for pertussis or influenza during pregnancy [13].

The Omicron variants (BA.1, BA.1.1), first detected in late 2021, are marked by numerous spike protein mutations that facilitate antibody evasion and contribute to breakthrough infections [14]. To address the diminished effectiveness of standard two-dose vaccines against these variants, the administration of additional boosters encoding the historical (Wuhan-1) spike protein was recommended for prevention of BA.1 [15]. By 2022, the emergence of the Omicron variant and its subvariants (BA.2, BA.2.12.1, BA.4, and BA.5) resulted in diminished effectiveness of the vaccines and boosters [16]. Bivalent boosters, which encode for the Wuhan-1 and either BA.1 or BA.4/5 spike antigens, are designed to boost protection against diverse Omicron variants [17]. Currently available bivalent COVID-19 vaccines included Comirnaty and Spikevax, introduced by Pfizer–BioNTech and Moderna, respectively [18,19]. In comparison to the original mRNA-1273 vaccine, they have demonstrated the ability to provoke heightened Nab responses against a variety of viral strains [18].

According to the previous literature, a COVID booster vaccine administered during pregnancy can increase the amount of Nab in the maternal blood and the transplacental antibodies in the umbilical cord blood compared to a two-dose primary vaccination [20]. Furthermore, the transplacental ratio against the Omicron BA.1 variant can reach 1.79–2.36 [20]. Regarding the bivalent COVID-19 vaccine, there are currently no studies reporting on the effects of transplacental antibody transfer in vaccinated pregnant women and their babies, and the reports on the effects of bivalent COVID-19 vaccines for Omicron XBB.1.5 subvariants are few. Hence, in our present study, we aim to not only review Nab inhibition in maternal blood and neonatal cord blood following bivalent COVID-19 vaccine administration, but also to evaluate the neutralization generated from bivalent COVID-19 vaccine against the newer Omicron subvariants, including BA.5, BF.7, BQ.1, and XBB.1.5.

## 2. Materials and Methods

### 2.1. Study Design

This prospective study was conducted at Kaohsiung Medical University Hospital under the auspices of the local institutional review board (IRB) (IRB number: KMUHIRB-SV(II)-20210087). The requirement of participant enrollment was as follows. All participants were confirmed singleton pregnancies and were enrolled during their hospital admission prior to delivery. The study expressly excluded patients experiencing preterm labor or gestational diabetes. All participants were at least 20 years old and exhibited no COVID-19 symptoms. Furthermore, they had no medical history requiring immunosuppressant treatment.

The participants in this study had all received three doses of COVID-19 vaccine before their current pregnancies, including a combination of zero to two doses of the Oxford/AstraZeneca ChAdOx1 nCoV-19 (AZD1222) vaccine paired with mRNA vaccines, such as the Pfizer BioNTech (BNT162b2) COVID-19 vaccine or the Spikevax (elasomeran) COVID-19 vaccine (previous called the mRNA-1273 Moderna vaccine). All participants received a fourth dose of vaccine, the Moderna COVID-19 bivalent (SPIKEVAX Bivalent Original/Omicron BA.1 or BA.4/5) vaccine that was administered during pregnancy. No significant discomfort was reported post-vaccination. Furthermore, study participants were also permitted to receive other regular vaccinations during the antenatal period, such as the tetanus toxoid, reduced diphtheria toxoid, and acellular pertussis (Tdap) vaccines (Adacel, Sanofi Pasteur, Toronto, ON, Canada), or the influenza (Flu) vaccine (AdimFlu-S, QIS, Adimmune Corporation, Taichung, Taiwan; FlucelvaxQuad, CSL Behring GmbH, Marburg, Germany; VAXIGRIP TETRA, Sanofi Pasteur, Val-de-Reuil, CEDEX, France), according to physicians or their individual preferences.

### 2.2. The Study Scheme for Collection of Samples and Clinical Data

In our study, we enrolled pregnant women who met specified eligibility criteria as of their delivery day and after the informed consent was signed. On the day of delivery, we collected both maternal peripheral blood samples and neonatal blood samples from the umbilical cord after cord-clamping. Both samples were sent to the laboratory for further examination. Additionally, the related clinical data were obtained from the electronic medical record system for statistical scrutiny. These data contained maternal age, parity, body mass index (BMI), as well as neonatal weight, gender, dates of COVID-19 vaccine receipt, and information on Tdap, flu vaccinations, time between vaccinations, and gestational weeks at delivery, among other relevant information. All the above data were collected for further analysis.

### 2.3. Neutralizing Antibody (Nab) Inhibition Test of SARS-CoV-2 Omicron BA.5, BF.7, BQ.1, and XBB.1.5 Variants

According to the study scheme, shown in Figure 1, we used an enzyme-linked immunosorbent assay (ELISA) process to determine Nab inhibition rate in response to various SARS-CoV-2 Omicron variants. Competitive binding between the spike protein receptor binding domain (SRBD), Nab, and angiotensin-converting enzyme (ACE) was instrumental in this process, as the resultant colorimetric alterations provided insights into Nab titers. We utilized commercially available ELISA kits (Cat. No. GTX538288) and followed all manufacturer instructions. We introduced samples, as well as positive and negative controls, into the wells of 96-well microplates. This was followed by the addition of a solution containing RBD proteins designed for different SARS-CoV-2 variants. These preparations were incubated at room temperature for one and half hours. Following incubation, the supernatant was discarded, and the wells were washed five times with a wash buffer. We then introduced a fresh conjugate solution and continued the incubation under similar conditions for an hour. We repeated the washing step and then added TMB solution to each well and incubated the plate in the dark for 15 min at room temperature. The reaction was terminated by the addition of a stop solution, leading to a visible color transition from blue to yellow.

The colorimetric shift was measured and quantified using a microplate spectrophotometer (Molecular Devices, San Jose, CA. USA) that read the O.D. value absorbance at 450 nm in each well. We used the gathered O.D values to determine the percentage of Nab inhibition, based on the application of the following formula:$$\mathrm{Inhibition}\;\%=\left(1-\frac{OD450\;value\;of\;sample}{average\;OD450\;value\;of\;negative\;control}\right)*100\%$$

### 2.4. Statistics

In our study, we conducted an analysis of the transplacental transmission of neutralizing Nab protection from mothers to infants, using the cord to maternal ratio of Nab inhibition rate. This rate was determined by comparing inhibition rates from neonatal cord blood to those from maternal serum. Additionally, we scrutinized the Nab inhibition rates produced by different subgroup populations including the intervals between the 4th COVID-19 vaccine to delivery, the intervals between the 3rd and 4th COVID-19 vaccines, previous monovalent COVID-19 vaccines combinations, maternal age, maternal BMI, neonatal body weight, and neonatal genders. We used non-parametric Mann–Whitney U test and ANOVA (analysis of variance) for statistical analyses. All data were processed using Microsoft Excel (Microsoft, Redmond, WA, USA) and SPSS Statistics (version 27, IBM, Armonk, NY, USA), with a p-value of less than 0.05 serving as the benchmark for statistical significance. We also illustrated these statistical findings by the software GraphPad Prism (version 9, GraphPad Software, San Diego, CA, USA) and BioRender (BioRender, Toronto, ON, Canada).

## 3. Results

### 3.1. Participants Characteristics

We gathered data from 11 pregnant women who met the selection criteria and were willing to participate in the research during March and April of 2023. Thus, a total of 11 maternal blood and 11 umbilical cord blood samples were collected. Table 1 presents the participants’ characteristics. The median age was 33 years (range 27–43), and the median maternal BMI was 28.12 (range 21.64–29.94). A total of 72.72% of the participants had one or two previous parities. Additionally, 72.72% of all participants delivered their child at 39 or 40 gestational weeks. The median newborn weight was 3150 g (range 2520–3555), and there were eight male and three female babies.

Four participants received the bivalent COVID-19 booster vaccine within the four weeks before childbirth, and two of them received vaccination during the week of delivery. Furthermore, four had a vaccination interval of 5–8 weeks, one had a vaccination interval of 9–12 weeks, and two had vaccination intervals exceeding 12 weeks. All participants received the first three doses of the vaccine prior to pregnancy, and five participants had the interval between the third dose and the bivalent COVID-19 vaccines as 41–44 weeks. Only one participant received antenatal vaccination with Tdap, and two received antenatal influenza vaccinations. Regarding the first three doses of the COVID-19 vaccine, five participants never received the AZD1222 vaccine, relying primarily on the mRNA vaccines. Among the remaining participants, five completed two full doses of the AZD1222 vaccine, while one had only received a single dose of the AZD1222 vaccine.

### 3.2. Neutralizing Antibody Inhibition Rates

Our research indicates that our participants demonstrated impressive immunity against different Omicron subvariants, with Nab inhibition rates of over 89% in maternal blood and 82% in neonatal cord blood. Additionally, the inhibition rates were 90.57% and 83.69%, even for the newer Omicron subvariant XBB.1.5. Furthermore, the ratio of Nab inhibition rates across different Omicron subvariants in both maternal and neonatal blood consistently surpassed 0.9, illustrating the significant transplacental transmission of Nab protection conferred by the bivalent COVID-19 vaccine. This compelling evidence attests to the robust Nab protection of the bivalent COVID-19 vaccines.

However, we identified two participants who were vaccinated with the bivalent COVID-19 vaccine within 1 week of childbirth that demonstrated no detectable Nab inhibition against any Omicron subvariants in either maternal or cord blood samples. We identified these two cases as outliers and excluded them from our subsequent analyses. The related data is displayed in Appendix A and Figure 2, and notes that there were no significant differences in Nab inhibition rates or their maternal-to-cord blood ratios for all Omicron subvariants studied.

### 3.3. Impact of Interval between Bivalent COVID-19 Vaccine and Childbirth to Neutralizing Antibody Inhibition Rates

As presented in Table 2, we evaluated the impact of varying time intervals—4, 8, and 12 weeks—between the administration of the bivalent COVID-19 vaccine and childbirth. We found there were no significant differences in the Nab inhibition rates for all Omicron subvariants studied. As previously noted, two participants in our study vaccinated during the week of delivery did not exhibit any Nab inhibition for all Omicron subvariants studied.

### 3.4. Impact of Interval between the Third Dose COVID-19 Vaccine and Spikevax Bivalent Vaccine on Neutralizing Antibody Inhibition Rates

Table 3 presents the effects of the time interval between the third dose of the COVID-19 vaccine and receipt of the bivalent COVID-19 vaccine booster administered during pregnancy on Nab inhibition in both maternal and umbilical cord blood against the Omicron subvariants studied. Our study found that there were only two participants with vaccination time intervals of less than 40 weeks and over 48 weeks between vaccinations, and most participants had an interval of approximately 40 to 48 weeks. However, regardless of whether this interval surpassed 40, 44, or 48 weeks, we observed no significant differences in Nab inhibition compared to those participants experiencing vaccination intervals less than 40, 44, or 48 weeks.

### 3.5. Impact of Previous COVID-19 Vaccine Combinations to Neutralizing Antibody Inhibition Rates

As shown in Table 4 and Figure 3, the type of previous COVID-19 vaccine received distinctly impacted Nab inhibition. We observed that participants who received three doses of mRNA vaccines (the BNT162b2 or mRNA-1273 Moderna vaccine) and those who did not receive the Oxford/AstraZeneca ChAdOx1 nCoV-19 vaccine (AZD1222) demonstrated higher Nab inhibition against the Omicron subvariant BA.5, BQ.1, and XBB.1.5 in maternal blood compared to those who received the AZD1222 vaccine (BA.5: 97.63% vs. 80.94%, *p* = 0.016; BQ.1: 96.57% vs. 84.11%, *p* = 0.032; XBB.1.5: 95.88% vs. 86.39%, *p* = 0.032). No difference was observed against Omicron subvariant BF.7, and there was no difference in the Nab inhibition for umbilical cord blood or the cord blood to maternal blood inhibition ratio.

This trend was also observed for those who had received two full doses of AZD1222 vaccines. Nab inhibition in maternal blood against Omicron subvariants BA.5, BF.7, and BQ.1 was lower in these participants compared to those who received one or no AZD1222 vaccine previously (BA.5: 78.95% vs. 88.91% vs. 97.63%, *p* = 0.046; BF.7: 94.28% vs. 98.06% vs. 97.50%, *p* = 0.088; BQ.1: 81.22% vs. 95.63% vs. 96.57%, *p* = 0.046). Furthermore, those who received two doses of AZD1222 vaccines also exhibited lower Nab inhibition in maternal blood against Omicron subvariants BA.5, BF.7, and BQ.1 compared to other vaccine combinations (BA.5: 78.95% vs. 95.88%, *p* = 0.032; BF.7: 94.28% vs. 97.61%, *p* = 0.032; BQ.1: 81.22% vs. 96.38%, *p* = 0.016). However, regardless of the COVID-19 vaccine combination received, there were no significant differences in Nab inhibition in the corresponding neonatal umbilical cord blood or the cord to maternal ratio.

### 3.6. Maternal and Neonatal Clinical Factors and Neutralizing Antibody Inhibition Rates

Table 5 and Table 6 present other potential clinical factors that might be thought to influence Nab inhibition in maternal blood and neonatal cord blood, including maternal age, maternal BMI, neonatal birth weight, and newborn gender. Our findings suggest that maternal age and BMI did not significantly affect Nab inhibition in maternal or cord blood against Omicron subvariants. Similarly, newborn gender demonstrated no effect on Nab inhibition in maternal or cord blood. Furthermore, we observed that Nab inhibition in maternal blood against Omicron subvariants BA.5, BF.7, and BQ.1 was higher when neonatal body weight was below 3000 or 3100 g (BA.5: 97.64% vs. 83.72%, *p* = 0.095; BF.7: 97.99% vs. 95.20%, *p* = 0.048; BQ.1: 96.67% vs. 86.13%, *p* = 0.095). However, there were no differences found for the subvariants XBB.1.5, and no significant difference in Nab inhibition in newborn cord blood or the cord to maternal ratio was observed.

## 4. Discussion

### 4.1. The Effectiveness of Bivalent COVID-19 Vaccination

Since the initial appearance of SARS-CoV-2 virus in 2019, the virus has undergone numerous transformations, leading to the emergence of several globally distributed variants of concern (VOCs) [21]. One such variant, Omicron (B.1.1.529), first reported in South Africa in 2021 [22], stands out due to numerous mutations in the receptor-binding domain (RBD) and the furin cleavage site [23]. These alterations resulted in higher binding affinity to the ACE-2 receptor, which may heighten its infectious capacity and modify its pathogenicity [24,25]. Models predict that Omicron BA.1 could be 10 times and 2.8 times more infectious than Wuhan-Hu-1 and the Delta variant, respectively, which is supported by in vitro studies and observations from epidemiological surveys [25]. For Omicron BA.5 and the related sublineage, including BQ.1 and BF.7, the genomic mutation and spike protein transformation can increase virus transmissibility and immune escape leading to higher risk of reinfection, even in people vaccinated with traditional COVID-19 vaccines [26,27]. The Omicron XBB1.5 subvariant has additional substitutions in SRBD resulting in more immune evasion capabilities than the above subvariants [27]. The bivalent COVID-19 vaccines containing mRNA encoding the spike protein from the Wuhan strain and BA.4/5 Omicron subvariants can provide significant additional protection against severe Omicron infection as booster doses have demonstrated an effectiveness of 58.7% compared to 25.2% for the monovalent booster [28]. However, the effectiveness of bivalent COVID-19 vaccine administration for pregnant women and newborns was not previously reported.

### 4.2. The Bivalent COVID-19 Vaccination for Pregnant Women and the Neonates

As we noted, the literature has demonstrated that antibodies generated in pregnant women following administration of conventional COVID-19 vaccines can be transmitted to newborns through transplacental transmission, providing them with Nab protection against wildtype, Alpha (B.1.1.7), Beta (B.1.351), Gamma (P.1), Delta (B.1.617.2), and Omicron (B.1.1.529) variants of SARS-CoV-2 [12,13]. In our current research, we examined Nab protection in pregnant women and their newborns following bivalent COVID-19 vaccination against the various SARS-CoV-2 Omicron subvariants, including BA.5, BF.7, BQ.1, and XBB.1.5. As with conventional COVID-19 vaccines, we found that bivalent COVID-19 vaccines offered substantial Nab inhibition against Omicron subvariants in maternal blood. The Nab protection can be also detected in cord blood of newborns, with the Nab inhibition rate exceeding 80%, indicating robust protection. Moreover, the ratio of Nab inhibition in cord blood-to-maternal blood was over 0.9. This can be compared to previous studies on conventional COVID-19 vaccines, in which the cord blood-to-maternal blood ratio for Nab inhibition ranged from 0.8% to 1.0% [12,29]. We can conclude, then, that pregnant women vaccinated with the bivalent COVID-19 vaccine can transmit Nab inhibition and provide significant protection against SARS-CoV-2 Omicron subvariants.

Per the examined literature, bivalent COVID-19 vaccine generated a stronger Nab titer and response than monovalent booster at 28 days after vaccination [30]. Compared to the monovalent booster, the binding antibody responses produced from bivalent vaccine were much stronger not only for early strains of SARS-CoV-2 such as Alpha, Beta, Gamma, and Delta, but also for Omicron BA.4/5 subvariants [18]. Previous research also demonstrated that Nab concentration began to increase as early as day 7 after vaccination and remained stable beyond 28 days [31]. Furthermore, some studies have also shown that the Nab titer and its seroresponse rate generated by bivalent COVID-10 vaccine can remain strong as long as 180 days after vaccination [17]. Clinical research has noted that the effectiveness of the bivalent vaccine booster for combating severe infections from multiple Omicron subvariants, including BQ.1 and XBB.1.5, at 2 weeks post vaccination was 67.4%, which changed to 38.4% at 20 weeks post vaccination [32]. In our study, we confirmed that pregnant women vaccinated less than a week before giving birth did not show detectable Nab inhibition against Omicron subvariants in maternal or cord blood. However, in cases where there is at least a four-week interval following vaccination, our study found that Nab inhibition was sufficient, with no significant differences in protection associated to varying intervals between vaccination and childbirth.

### 4.3. Different Factors Related to Neutralizing Antibody Inhibition after Bivalent COVID-19 Vaccination in Pregnant Women

Although there is variation of vaccination strategies among different countries [33], according to current vaccination guidelines for individuals aged 12 and above in the United States and the World Health Organization, there should be at least an 8-week interval between the administration of the monovalent mRNA COVID-19 vaccine and the subsequent mRNA bivalent COVID-19 vaccine [34]. The time strategy can assure vaccine efficacy and reduce the risks of myocarditis and pericarditis associated with vaccination. In our study, all participants received the bivalent COVID-19 vaccine between 36 to 52 weeks following receipt of the monovalent COVID-19 vaccine. Because the intervals were much longer than 8 weeks, we did not observe significant differences in Nab inhibition in maternal blood or cord blood for different intervals, including 40-, 44-, or 48-week intervals examined in our study.

We also studied the impact of different combinations of previous monovalent COVID-19 vaccines on the immunogenicity of the bivalent COVID-19 vaccine. Historically, a heterologous monovalent COVID-19 vaccination regimen involving combinations such as AZD1222 and mRNA vaccines such BNT161b2 has been shown to yield higher levels of spike RBD antibodies [35]. Concurrently, compared to homologous AZD1222 vaccination, the heterologous vaccination has demonstrated a stronger immune response, including higher levels of spike specific CD4+ and CD8+ T cells [36]. Other studies have found that substantial Nab protection can be achieved when administration of the bivalent COVID-19 vaccine follows monovalent vaccinations containing either homologous mRNA vaccines or heterologous mRNA/adenovirus-based vaccines [31]. In our study, pregnant women who previously received one or two doses of AZD1222 vaccines and then received the bivalent COVID-19 vaccine demonstrated weaker Nab inhibition against the Omicron subvariants studied compared to those who received only the monovalent mRNA COVID-19 vaccines. Because the bivalent COVID-19 vaccines are also mRNA vaccines, they were expected to elicit stronger immunity for those who previously received monovalent mRNA vaccines compared to AZD1222 vaccinations. However, such differences were not found in neonatal cord blood. Regardless of the type of monovalent vaccination previously received by the mother, receipt of the bivalent COVID-19 vaccine conferred protection to the newborn in a uniform manner.

Regarding other factors relating to pregnant women and their newborns, an interesting observation from our study was that mothers of lower birth weight newborns exhibited stronger Nab inhibition against the Omicron subvariants studied. However, regardless of newborn weight, there were no significant differences in cord blood Nab inhibition. This was echoed in the cord-to-maternal blood ratio of Nab inhibition. This suggests that lower newborn body weights may be less of a demand or draw less from the mother’s level of protection, which may be beneficial to the mother. However, there were no similar correlations associated to the mother’s age or BMI. Additional data could provide a more comprehensive explanation for any potential correlated effects associated to body weight and immunity transference.

### 4.4. Clinical Significance of Our Current Study

Our findings indicate that the bivalent COVID-19 vaccine delivers strong Nab inhibition against multiple SARS-CoV-2 Omicron subvariants in both maternal and cord blood. We also observed that previous vaccination with AZD1222 vaccines, monovalent vaccination strategies, and higher neonatal birth weight reduced maternal Nab inhibition. However, Nab inhibition in newborn cord blood demonstrated no significant inhibition differences irrespective of these variables. Hence, administering the bivalent COVID-19 vaccine to pregnant women remains an effective preventive measure in the evolving pandemic landscape abundant with subvariants.

### 4.5. Limitation of Our Current Study

To the best of our knowledge, our study is the first to investigate the impact of bivalent COVID-19 vaccine on maternal and newborn cord blood, and the associated transmission ratio between mother and child. It is also one of the few studies to research bivalent COVID-19 vaccine efficacy against the recent SARS-CoV-2 Omicron subvariants, including the XBB.1.5 subtype. This study did have limitations, the most significant of which was the small number of participants, which resulted in a limited sample size, making it challenging to conduct subgroup analysis for different variates. No unvaccinated populations can be included since the promotion of COVID-19 vaccines for pregnant women currently. In addition, pregnant women’s immunity status may also be related to proteins, physical activity, and nutrition that were not demonstrated in our study owing to the small sample size [37]. In our participant pool, two cases received their vaccination during the week of delivery, resulting in undetectable Nab inhibition rates, which further reduced our case count. Therefore, more participants will be needed in future studies to strengthen and validate our findings.

## 5. Conclusions

Our study revealed that administration of the bivalent COVID-19 vaccine in pregnant women provides robust neutralizing inhibition, in both maternal and newborn bodies, against the new SARS-CoV-2 Omicron subvariants, including BA.5, BF.7, BQ.1, and XBB.1.5. If administered less than a week before delivery, the vaccine’s protective power may be insufficient. Furthermore, our findings indicate that those who’s monovalent COVID-19 vaccination contained AZD1222 vaccines and those with heavier newborns may experience slightly weaker Nab inhibition in maternal blood, although the level of protection remains above 40%. Thus, our study suggests that the bivalent COVID-19 vaccine indeed provides effective protection against various Omicron subvariants. However, a larger sample size is necessary for further validation.

## Figures and Tables

**Figure 1 vaccines-11-01425-f001:**
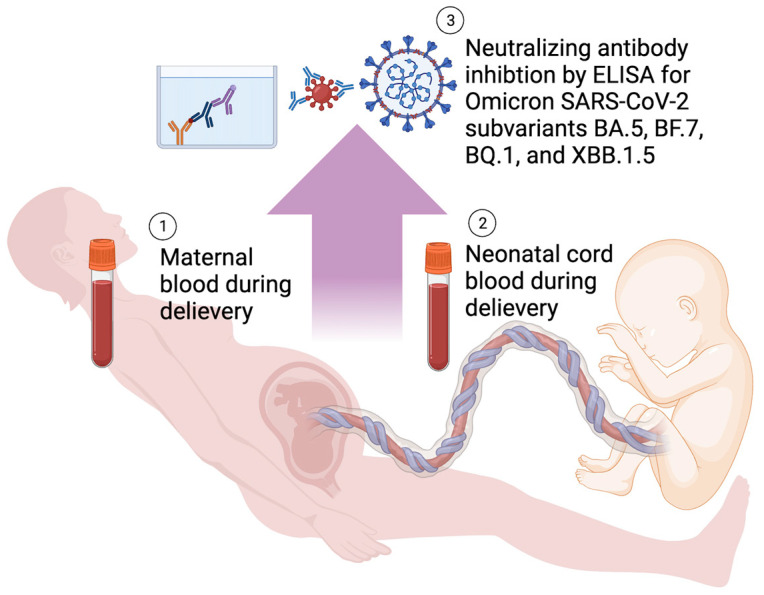
Study Scheme. ELISA, enzyme-linked immunosorbent assay.

**Figure 2 vaccines-11-01425-f002:**
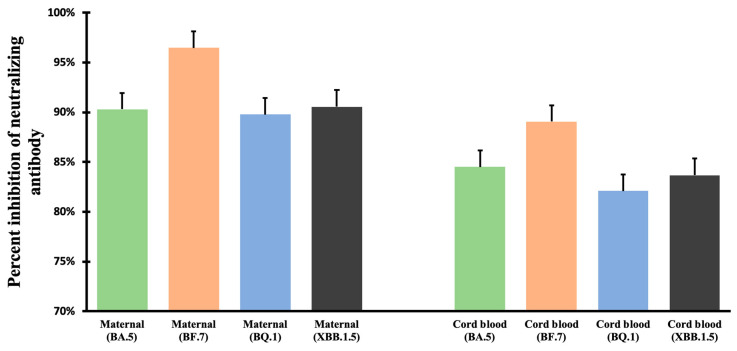
Neutralizing antibody inhibition in maternal and cord blood for Omicron BA.5, BF.7, BQ.1, and XBB.1.5 SARS-CoV-2.

**Figure 3 vaccines-11-01425-f003:**
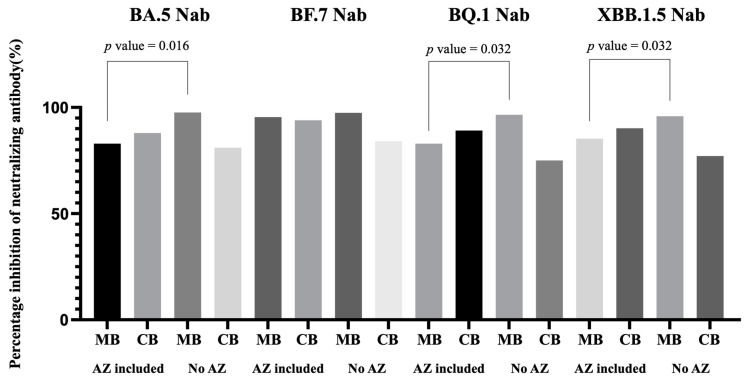
Previous monovalent COVID-19 vaccine combinations containing AZD1222 vaccine or not for neutralizing antibody inhibition in maternal and cord blood for Omicron BA.5, BF.7, BQ.1, and XBB.1.5 SARS-CoV-2. Nab, neutralizing antibody; MB, maternal blood; CB, neonatal cord blood; AZ, AZD1222 vaccine.

**Table 1 vaccines-11-01425-t001:** Participants characteristics.

	N (Total N = 11)	Percentage (%)
Median age, years (range)	33 (27–43)	
Previous parity		
0	2	18.18
1	4	36.36
2	4	36.36
3	1	9.09
Median BMI (range)	28.12 (21.64–29.94)	
Delivery weeks		
38 weeks	3	27.27
39 weeks	4	36.36
40 weeks	4	36.36
Interval to delivery		
0–4 weeks	4	36.36
5–8 weeks	4	36.36
9–12 weeks	1	9.09
>12 weeks	2	18.18
Interval between 3rd and 4th COVID vaccine		
36–40 weeks	2	18.18
41–44 weeks	5	45.45
45–48 weeks	2	18.18
>48 weeks	2	18.18
Tdap vaccine		
Yes	10	90.90
No	1	9.09
Flu vaccine		
Yes	9	81.81
No	2	18.18
Previous COVID vaccine		
2 AZ before	5	45.45
1 AZ before	1	9.09
No AZ before	5	45.45
Median neonatal body weight, gm (range)	3150 (2520–3555)	
Neonatal gender		
Male	8	72.73
Female	3	27.27

BMI, body mass index; Tdap vaccine, tetanus toxoid, reduced diphtheria toxoid, and acellular pertussis vaccine; Flu vaccine, influenza vaccine. AZ, AZD1222 (the Oxford/AstraZeneca ChAdOx1 nCoV-19) vaccine.

**Table 2 vaccines-11-01425-t002:** The impacts of intervals between bivalent COVID-19 booster vaccine and baby delivery on neutralizing antibody inhibition in maternal and cord blood for Omicron BA.5, BF.7, BQ.1, and XBB.1.5 SARS-CoV-2.

Interval	N	BA.5			BF.7			BQ.1			XBB.1.5		
		MB (%)	CB (%)	Ratio	MB (%)	CB (%)	Ratio	MB (%)	CB (%)	Ratio	MB (%)	CB (%)	Ratio
<4 weeks	4 ^a^	95.58	62.51	0.66	97.89	68.20	0.69	88.55	45.84	0.54	94.44	53.55	0.56
≥4 weeks	7 ^b^	86.30	91.85	1.04	95.62	96.02	1.00	89.96	94.19	1.06	89.51	93.73	1.06
*p* value		0.558	0.286	0.643	0.500	0.143	0.071	0.500	0.143	0.143	0.889	0.143	0.071
<8 weeks	8 ^c^	88.87	82.72	0.91	96.15	85.52	0.88	87.20	73.61	0.87	90.39	78.59	0.88
≥8 weeks	3	87.34	87.52	1.00	96.08	94.97	0.980	94.54	96.25	1.02	91.02	92.18	1.01
*p* value		0.905	0.571	0.393	0.905	1.000	0.786	0.262	0.393	1.000	0.548	0.786	1.000
<12 weeks	9 ^c^	90.04	85.20	0.93	96.39	87.48	0.90	88.72	77.29	0.89	91.19	81.42	0.90
≥12 weeks	2	82.47	82.47	1.00	95.22	93.84	0.98	92.87	96.54	1.04	88.57	90.47	1.02
*p* value		1.000	0.857	0.286	1.000	1.000	0.857	0.667	0.429	0.643	0.889	0.857	0.857

MB, maternal blood; CB, cord blood. ^a^ 2 cases were not included due to the inability to detect any neutralizing antibody inhibition. ^b^ 1 case had no calculated values for cord blood due to the lack of an available sample. ^c^ 2 cases were not included due to the inability to detect any neutralizing antibody inhibition. Further, 1 case omitted cord blood values because there was no available sample.

**Table 3 vaccines-11-01425-t003:** The impacts of intervals between previous monovalent COVID-19 vaccine administration, bivalent COVID-19 vaccine administration, and delivery date on neutralizing antibody inhibition in maternal and cord blood for Omicron BA.5, BF.7, BQ.1, and XBB.1.5 SARS-CoV-2.

Interval	N	BA.5			BF.7			BQ.1			XBB.1.5		
		MB (%)	CB (%)	Ratio	MB (%)	CB (%)	Ratio	MB (%)	CB (%)	Ratio	MB (%)	CB (%)	Ratio
≤40 weeks	2	85.78	95.35	1.11	95.84	96.76	1.01	80.57	89.33	1.13	83.12	94.47	1.15
>40 weeks	9 ^a^	89.09	80.91	0.89	96.21	86.50	0.90	92.24	79.69	0.86	92.75	80.09	0.86
*p* value		0.500	0.857	0.071	1.000	0.429	0.143	0.500	0.857	0.143	0.222	0.643	0.071
≤44 weeks	7 ^b^	86.67	77.15	0.91	95.62	84.94	0.89	87.81	75.97	0.89	87.12	77.38	0.91
>44 weeks	4 ^c^	90.47	96.79	1.00	96.76	95.95	0.98	91.94	92.32	0.99	94.96	94.19	0.98
*p* value		0.730	0.250	1.000	0.730	0.780	0.571	0.556	0.571	0.250	0.286	0.571	0.250
≤48 weeks	9 ^b^	89.32	82.65	0.94	96.27	87.90	0.91	88.61	80.16	0.92	89.81	81.98	0.93
>48 weeks	2 ^c^	84.99	97.61	1.01	95.62	97.25	0.99	93.28	95.67	0.98	93.41	95.59	1.00
*p* value		0.667	0.750	1.000	0.667	1.000	1.000	0.667	1.000	0.500	1.000	1.000	0.750

MB, maternal blood; CB, cord blood. ^a^ 2 cases were not included due to the inability to detect any neutralizing antibody inhibition. Further, 1 case omitted cord blood values because there was no available sample. ^b^ 2 cases were not included due to the inability to detect any neutralizing antibody inhibition. ^c^ 1 case had no calculated values for cord blood due to the lack of an available sample.

**Table 4 vaccines-11-01425-t004:** The impacts of previous monovalent COVID-19 vaccine combinations on neutralizing antibody inhibition in maternal and cord blood for Omicron BA.5, BF.7, BQ.1, and XBB.1.5 SARS-CoV-2.

Vaccines	N	BA.5			BF.7			BQ.1			XBB.1.5		
		MB (%)	CB (%)	Ratio	MB (%)	CB (%)	Ratio	MB (%)	CB (%)	Ratio	MB (%)	CB (%)	Ratio
AZ included	6 ^a^	80.94	87.97	1.06	95.03	93.98	0.98	84.11	89.14	1.09	86.39	90.22	1.07
No AZ	5 ^b^	97.63	81.06	0.83	97.50	84.16	0.86	96.57	75.06	0.77	95.88	77.15	0.79
*p* value		0.016	0.343	0.114	0.111	0.886	0.886	0.032	0.486	0.114	0.032	0.686	0.200
2 AZ	5 ^a^	78.95	85.11	1.05	94.28	92.62	0.98	81.22	86.34	1.11	84.66	87.67	1.08
1 AZ	1	88.91	96.55	1.09	98.06	98.05	1.00	95.63	97.53	1.02	93.30	97.90	1.05
No AZ	5 ^b^	97.63	81.06	0.83	97.50	84.16	0.86	96.57	75.06	0.77	95.88	77.15	0.80
*p* value		0.046	0.400	0.200	0.088	0.201	0.806	0.046	0.392	0.223	0.083	0.311	0.297
2 AZ	5 ^a^	78.95	85.11	1.05	94.28	92.62	0.98	81.22	86.34	1.11	84.66	87.66	1.08
1 or no AZ	6 ^b^	95.88	84.16	0.88	97.61	86.94	0.89	96.38	79.55	0.82	95.36	81.30	0.85
*p* value		0.032	0.250	0.393	0.032	0.250	1.000	0.016	0.250	0.250	0.063	0.250	0.571

MB, maternal blood; CB, cord blood; AZ, AZD1222 (the Oxford/AstraZeneca ChAdOx1 nCoV-19) vaccine. ^a^ 1 case was not included due to the inability to detect any neutralizing antibody inhibition. Further, 1 case omitted cord blood values because there was no available sample. ^b^ 1 case was not included due to the inability to detect any neutralizing antibody inhibition.

**Table 5 vaccines-11-01425-t005:** The impacts of maternal age and body mass index on neutralizing antibody inhibition in maternal and cord blood for Omicron BA.5, BF.7, BQ.1, and XBB.1.5 SARS-CoV-2.

Maternal Condition	N	BA.5			BF.7			BQ.1			XBB.1.5		
		MB (%)	CB (%)	Ratio	MB (%)	CB (%)	Ratio	MB (%)	CB (%)	Ratio	MB (%)	CB (%)	Ratio
Age < 34	6 ^a^	91.97	96.13	1.05	96.64	96.23	1.00	87.67	91.02	1.05	90.78	94.09	1.05
Age ≥ 34	5 ^b^	83.85	65.15	0.77	95.49	77.14	0.81	92.11	67.23	0.73	90.39	66.34	0.75
*p* value		0.905	0.571	0.036	1.000	0.393	0.250	0.905	1.000	0786	0.905	0.571	0.571
BMI < 28	5 ^b^	89.12	97.34	1.03	96.33	97.67	1.00	94.61	96.97	1.00	94.13	96.80	1.02
BMI ≥ 28	6 ^a^	87.75	76.82	0.89	95.97	83.91	0.88	85.67	73.18	0.88	87.79	75.81	0.88
*p* value		1.000	0.250	0.571	0.905	0.143	0.393	0.286	0.250	1.000	0.730	0.250	0.786

MB, maternal blood; CB, cord blood; BMI, body mass index. ^a^ 1 case was not included due to the inability to detect any neutralizing antibody inhibition. ^b^ 1 case was not included due to the inability to detect any neutralizing antibody inhibition. Further, 1 case omitted cord blood values because there was no available sample.

**Table 6 vaccines-11-01425-t006:** The impacts of neonatal weight and gender on neutralizing antibody inhibition in maternal and cord blood for Omicron BA.5, BF.7, BQ.1, and XBB.1.5 SARS-CoV-2.

	N	BA.5			BF.7			BQ.1			XBB.1.5		
		MB (%)	CB (%)	Ratio	MB (%)	CB (%)	Ratio	MB (%)	CB (%)	Ratio	MB (%)	CB (%)	Ratio
NBW < 3000 g	3	97.64	75.46	0.77	97.99	79.65	0.81	96.67	67.51	0.69	95.71	70.56	0.73
NBW ≥ 3000 g	8 ^a^	83.72	89.95	1.05	95.20	94.72	0.99	86.13	90.86	1.07	88.06	91.56	1.06
*p* value		0.037	0.590	0.354	0.036	0.490	0.433	0.165	0.512	0.330	0.112	0.508	0.350
NBW < 3100 g	5 ^b^	97.64	75.46	0.77	97.99	79.65	0.81	96.67	67.51	0.69	95.71	70.56	0.73
NBW ≥ 3100 g	6 ^c^	83.72	89.95	1.05	95.20	94.72	0.99	86.13	90.86	1.07	88.06	91.56	1.06
*p* value		0.037	0.590	0.354	0.036	0.490	0.433	0.165	0.512	0.330	0.112	0.508	0.350
NBW < 3200 g	6 ^b^	89.90	73.27	0.83	96.58	82.17	0.85	94.41	74.34	0.79	91.65	73.65	0.81
NBW ≥ 3200 g	5 ^c^	87.13	95.77	1.06	95.77	95.97	1.00	85.83	89.86	1.07	89.77	93.71	1.06
*p* value		0.752	0.255	0.238	0.641	0.365	0.368	0.244	0.514	0.288	0.768	0.366	0.292
Male baby	8 ^d^	89.09	80.91	0.89	96.21	86.50	0.90	92.24	79.69	0.86	92.75	80.09	0.86
Female baby	3 ^e^	85.78	95.35	1.11	95.84	96.76	1.01	80.57	89.33	1.13	83.12	94.47	1.15
*p* value		0.751	0.508	0.328	0.858	0.541	0.517	0.579	0.729	0.387	0.175	0.556	0.297

MB, maternal blood; CB, cord blood; NBW, neonatal body weight. ^a^ 2 cases were not included due to the inability to detect any neutralizing antibody inhibition. Further, 1 case omitted cord blood values because there was no available sample. ^b^ 2 cases were not included due to the inability to detect any neutralizing antibody inhibition. ^c^ 1 case had no calculated values for cord blood due to the lack of an available sample. ^d^ 1 case was not included due to the inability to detect any neutralizing antibody inhibition. Further, 1 case omitted cord blood values because there was no available sample. ^e^ 1 case was not included due to the inability to detect any neutralizing antibody inhibition.

## Data Availability

Not applicable.

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
