# Peer review of "COVID-19 Bivalent Booster in Pregnancy: Maternal and Neonatal Antibody Response to Omicron BA.5, BQ.1, BF.7 and XBB.1.5 SARS-CoV-2"

_vaccines, 2023, doi:10.3390/vaccines11091425_

Round 1
Reviewer 1 Report
Chen et al investigated the effects of bivalent COVID-19 booster vaccination during pregnancy on Nab levels in maternal blood, transplacental transmission in umbilical cord blood (CB), and efficacy against Omicron SARS-CoV-2 subvariants including 29 BA.5, BF.7, BQ.1, and XBB.1.5. The story sound interesting and the manuscript is well-organized. I think it is can be acceptable in present form.
Author Response
Chen et al investigated the effects of bivalent COVID-19 booster vaccination during pregnancy on Nab levels in maternal blood, transplacental transmission in umbilical cord blood (CB), and efficacy against Omicron SARS-CoV-2 subvariants including 29 BA.5, BF.7, BQ.1, and XBB.1.5. The story sound interesting and the manuscript is well-organized. I think it is can be acceptable in present form.
Response: We are deeply grateful for the comments and suggestions you have provided.
Reviewer 2 Report
The study aimed to evaluate the antibody response after anti-covid vaccination during pregnancy.
General. The study is publishable, but needs significant improvement before acceptance.
I have no major issues, but the many minor issues require a major revision.
Minor issues.
1. Why the authors did not also include totally unvaccinated women to test for possible differences between the two cohorts of people?
2. The authors employed parametric tests in a small number of subjects. Were the data normally distributed? If no, then please redo the analyses by means of non-parametric tests.
3. Subsection 3.1. can be moved to appendix.
4. Figures 1. I suggest to colorise the histograms.
5. Please include some more graphs to allow readers to get a quick grasp of the results.
6. The discussion should be divided into subsection to improve flow of reading.
7. Please add a paragraph with clinical significance of the findings.
8. Please add further references in the discussion to allow better understanding of the global situation regarding vaccination of pregnant women.
After successful revision, the manuscript can be accepted.
Author Response
Why the authors did not also include totally unvaccinated women to test for possible differences between the two cohorts of people?
Response: Thank you for this comment. Due to the promotion of vaccination and the COVID-19 infection status, it is challenging to find population who either has not been vaccinated or infected in Taiwan. Moreover, pregnant women with COVID-19 infection are more likely to experience severe outcomes, making it even more challenging to find such a group. We have included this point in the limitation section of our revised manuscript, Line 421-423, as highlighted in yellow.
The authors employed parametric tests in a small number of subjects. Were the data normally distributed? If no, then please redo the analyses by means of non-parametric tests.
Response: Thank you for this valuable comment. Due to the limited number of cases, it is indeed hard to have a normal distribution for our data. We have re-analyzed the data using non-parametric statistics, re-organized the figures and tables, and re-checked the writing in our revised manuscript, Line 159, Lines 163-165, Lines 250-253, Lines 272-273, Lines 276-277, Lines 286-289, as highlighted in yellow.
Subsection 3.1. can be moved to appendix.
Response: Thank you for this comment. The subsection 3.1 in our manuscript describes the basic status of these participants. The overall participation status is essential for understanding the study. Thank you for this suggestion, and we prefer to keep it in the main text for readers' reference.
Figures 1. I suggest to colorise the histograms.
Response: Thank you for this comment. We have prepared it as a colored version (Figure 2 in our revised manuscript).
Please include some more graphs to allow readers to get a quick grasp of the results.
Response: Thank you for this valuable comment. We have added Figure 1 and Figure 3 in our revised manuscript.
Figure 1. Study scheme.
Fgiure 3. Previous monovalent COVID-19 vaccine combinations containing AZD1222 vaccine or not for neutralizing antibody inhibition in maternal and cord blood for Omicron BA.5, BF.7, BQ.1, and XBB.1.5 SARS-CoV-2.
The discussion should be divided into subsection to improve flow of reading.
Response: Thank you for your suggestion! We have divided the entire section into different subsections to facilitate reading (see discussion section in our revised manuscript).
Please add a paragraph with clinical significance of the findings.
Response: Thank you for this comment. We have prepared it in our revised manuscript, Lines 406-413, as highlighted in yellow.
Please add further references in the discussion to allow better understanding of the global situation regarding vaccination of pregnant women.
Response: Thank you for this comment. We have added descriptions and references related to the vaccine policy, Lines 365-369, as highlighted in yellow.
Reviewer 3 Report
The manuscript submitted for publication to Vaccines by Chen et al., titled: "COVID-19 Bivalent Booster in Pregnancy: Maternal and Neo-natal Antibody Response to Omicron BA.5, BQ.1, BF.7 and XBB.1.5 SARS-CoV-2" in an interesting study aiming to investigate the effect of the booster in pregnancy. The study is well designed and well written. The reviewer would like to make some points below for the improvement of the manuscript that the authors could consider:
1. How was the number of participants determined?
2. What were the inclusion and exclusion criteria (consider specifying them clearly), this is important to consider for potential confounding factors such as medications, supplements etc.
3. Did the authors consider the effect of diet and physical activity? Diet is particularly especially certain aspect of it may extend immunoresponses. A brief discussion on the topic of diet and immunology would strengthen the paper. Here is a manuscript that may be useful:
- Sikalidis AK (2015) Amino Acids and Immune Response: A role for cysteine, glutamine, phenylalanine, tryptophan and arginine in T-cell function and cancer? Pathol Oncol Res. 21(1):9-17. doi: 10.1007/s12253-014-9860-0.
4. Was gestational diabetes considered in the analyses? If none of the participants developed gestational diabetes that should be mentioned.
English language is OK overall, the manuscript would benefit from a native English speaker proofread.
Author Response
How was the number of participants determined?
Response: Thank you for this comment. Given that there were not many pregnant women who received the fourth dose of either COVID-19 mRNA vaccine or bivalent COVID-19 mRNA vaccine, and the efficacy of mRNA vaccine against the newer subvariants of omicron SARS-CoV-2 in pregnant women remains unknown, we promptly have embarked on this research after obtaining samples from the current 11 pregnant women. We hope to provide a reference for frontline clinical care providers via the findings from our study.
What were the inclusion and exclusion criteria (consider specifying them clearly), this is important to consider for potential confounding factors such as medications, supplements etc.
Response: Thank you for this comment. In the subsection 2.1 of our manuscript, we have described the relevant qualified conditions for these participants as well as the conditions that were not met, that is, the so-called inclusion and exclusion criteria. If more clarity is desired, we are more than happy to further emphasize the text within our manuscript, Line 95, as highlighted in yellow.
Did the authors consider the effect of diet and physical activity? Diet is particularly especially certain aspect of it may extend immunoresponses. A brief discussion on the topic of diet and immunology would strengthen the paper. Here is a manuscript that may be useful:
Sikalidis AK (2015) Amino Acids and Immune Response: A role for cysteine, glutamine, phenylalanine, tryptophan and arginine in T-cell function and cancer? Pathol Oncol Res. 21(1):9-17. doi: 10.1007/s12253-014-9860-0.
Response: Thank you for this comment. Although it is slightly off-topic from our study, we have included this reference in our revised manuscript, Lines 423-425, as highlighted in yellow.
Was gestational diabetes considered in the analyses? If none of the participants developed gestational diabetes that should be mentioned.
Response: Thank you for this comment. None of the participants in our study had gestational diabetes, and we have clarified this point, Line 98, as highlighted in yellow.
Round 2
Reviewer 2 Report
The authors have addressed correctly all the concerns and have made extensive modifications in the manuscript to improve the scientific quality.
Subject to editing some linguistic slips scattered throughout the manuscript, the revised submission can be accepted.
Subject to editing some linguistic slips scattered throughout the manuscript, the revised submission can be accepted.
Author Response
The authors have addressed correctly all the concerns and have made extensive modifications in the manuscript to improve the scientific quality.
Subject to editing some linguistic slips scattered throughout the manuscript, the revised submission can be accepted.
Response: Thank you for this comment. We have checked and revised our revised manuscript again!
Reviewer 3 Report
The authors have made a reasonable effort in addressing the reviewer's points.
Author Response
The authors have made a reasonable effort in addressing the reviewer's points.
Response: Thank you for this comment!